# Machine Learning Methods and Visual Observations to Categorize Behavior of Grazing Cattle Using Accelerometer Signals

**DOI:** 10.3390/s24103171

**Published:** 2024-05-16

**Authors:** Ira Lloyd Parsons, Brandi B. Karisch, Amanda E. Stone, Stephen L. Webb, Durham A. Norman, Garrett M. Street

**Affiliations:** 1Quantitative Ecology and Spatial Technologies Laboratory, Department of Wildlife, Fisheries and Aquaculture, Mississippi State University, Starkville, MS 39762, USA; durham.norman21@gmail.com (D.A.N.); gms264@msstate.edu (G.M.S.); 2West River Research and Extension Center, Department of Animal Science, South Dakota State University, Rapid City, SD 57703, USA; 3Department of Animal and Dairy Sciences, Mississippi State University, Starkville, MS 39762, USA; brandi.karisch@msstate.edu (B.B.K.); amandaestone1230@gmail.com (A.E.S.); 4Texas A&M Natural Resources Institute and Department of Rangeland, Wildlife, and Fisheries Management, Texas A&M University, College Station, TX 77843, USA; stephen.webb@ag.tamu.edu

**Keywords:** accelerometers, activity budget, beef cattle, precision livestock technology, spatially explicit modeling, behavior landscapes

## Abstract

Accelerometers worn by animals produce distinct behavioral signatures, which can be classified accurately using machine learning methods such as random forest decision trees. The objective of this study was to identify accelerometer signal separation among parsimonious behaviors. We achieved this objective by (1) describing functional differences in accelerometer signals among discrete behaviors, (2) identifying the optimal window size for signal pre-processing, and (3) demonstrating the number of observations required to achieve the desired level of model accuracy,. Crossbred steers (Bos taurus indicus; *n* = 10) were fitted with GPS collars containing a video camera and tri-axial accelerometers (read-rate = 40 Hz). Distinct behaviors from accelerometer signals, particularly for grazing, were apparent because of the head-down posture. Increasing the smoothing window size to 10 s improved classification accuracy (*p* < 0.05), but reducing the number of observations below 50% resulted in a decrease in accuracy for all behaviors (*p* < 0.05). In-pasture observation increased accuracy and precision (0.05 and 0.08 percent, respectively) compared with animal-borne collar video observations.

## 1. Introduction

Precision livestock management is the new frontier in grazing livestock production [1]. The livestock production industry faces a series of challenges, such as labor shortages, social pressures, and increasing efforts to implement sustainable grazing practices to protect sensitive ecosystems [1,2,3,4]. These challenges may be overcome by developing smart tracking technology [4,5], such as global positioning systems (GPS) and accelerometers [1,6], temperature and heart-rate monitors [7], and video cameras [8]. Behavior classification from collars containing GPS and accelerometers represents a relatively common obstacle to behavior, spatial, and metabolic research within animal science, rangeland ecology, animal behavior, and the ecological literature over the past decade [9]. Improving behavioral categorization in cattle grazing extensive landscapes may be a key to improving precision livestock technology [10,11].

Monitoring animal behavior using wearable sensors is an extremely difficult endeavor. Indeed, the number of papers devoted to animal-borne sensors and accelerometers, as components of precision animal agriculture, has risen exponentially [12]. The development of transferable, parsimonious models with high degrees of accuracy across deployments is the gold standard for accelerometer-based behavior monitoring systems [13,14]. Machine learning methodologies provide extremely high precision and accuracy but require large amounts of data for training the models [14,15].

Machine learning methods offer the potential for making quick and accurate decisions once developed and promote an interdisciplinary approach to innovation. However, budding ethologists are confronted with a large number of potential pre-processing methods for analyzing accelerometer data to classify behavior [16,17]. Recent research indicates that smoothing windows has the largest impact on improving the accuracy of behavioral classification, with overlapping windows providing important improvements in datasets with limited data [17]. Concomitantly filtering provides varying results but often does not improve data quality [16,17]. Thus, it is important to consider both the pre-processing method applied to accelerometer data alongside the chosen machine learning tool, as well as the biological relevance and desired temporal resolution of behavioral output.

Observations of behavior, used to train machine learning classification models, are collected via direct observation or video recording. However, validation and training data collection is labor intensive, which often means the total data collected is minimal compared with the duration of the study period and short of what would be desired by big data scientists [15]. To overcome issues related to direct behavioral observations, wildlife ecologists have confronted this issue by using remote sensing technologies such as motion sensors, accelerometers, acoustic recorders, and animal-borne camera systems (camera collars) to classify animal behavior and activity across broad temporal and spatial scales at fine resolution [18,19]. Rangeland ecologists experience similar limitations when working in extensive rangelands [3,6,20]. Behavior classification has been collected via video cameras mounted on tracking collars [18,21,22,23]. However, no direct comparison has been conducted between direct in-pasture behavioral observation and the camera footage provided by animal-borne video cameras. Utilization of animal-borne video cameras to calibrate behavior identification algorithms provides several potential benefits, including increasing the quantity of behaviors observed and temporal representation of the animal’s behavioral ethogram.

### Objectives

The purpose of this paper was to collect observed behavior and pre-process accelerometer signals to differentiate parsimonious behaviors. To accomplish this, our objectives were to (1) describe functional differences in accelerometer signals between discrete behaviors, (2) identify the optimal smoothing window size for accelerometer signal pre-processing to maximize behavior classification accuracy, and (3) demonstrate the number of behavior observations required to achieve the desired level of trained model accuracy.

## 2. Methods

### 2.1. Animals and Equipment

All experiments were approved by the Mississippi State University Institutional Animal Care and Use Committee (IACUC-20502). This study was part of a broader research project that assessed fundamental grazing ecology principles on animal growth and fitness. Brahman steers (*Bos taurus indicus*; *n* = 10), with an initial body weight of 232 ± 32 kg, were grazed continuously on pasture from 21 February 2020 to 3 December 2020. The study was located at the Pasture Demonstration Farm (Ardmore, OK, USA), owned, and operated by the Noble Research Institute. We used a single paddock (±11.5 ha) throughout the entirety of this study, which was composed of bermudagrass (*Cynodon dactylon*) and tall fescue (*Festuca arundinacea*) and inter-seeded with winter annual ryegrass (*Secale cereale*) in November 2019. Water was provided via an automatic ball-type waterer (MiraFount, Miraco Livestock Water Systems, Grinnell, IA, USA) situated in the fence line between adjoining pastures [5,24].

Cattle were fitted with Vectronic Vertex Plus GPS collars (Vectronic Aerospace GmbH, Berlin, Germany) equipped with tri-axial accelerometers and magnetometers (Wildbytes Technologies Ltd., Swansea University, Swansea, UK) powered by 1 D-cell battery and a forward-facing video camera (Figure 1, Figure 2 and Figure 3) powered by 2 D-cell lithium-ion batteries. The collars weighed approximately 1500 g and were programmed to collect GPS location fix data at 15-min intervals, and the forward-facing camera system recorded a 15-s video recording at 1-h intervals between sunrise and sunset. Both GPS location fixes and collar videos were timestamped according to Coordinated Universal Time (UTC). Accelerometers were programmed to record the magnitude of 3-dimensional acceleration at 40 Hz and magnetometry at 10 Hz. Accelerometers were synchronized with UTC and calibrated before deployment; they were replaced at three-month intervals, or as needed, for the duration of the study period. A housing was fashioned from a 1-1/4 inch schedule 40 PVC pipe, which contained a lithium battery and an insert fashioned from Kaizen foam to hold the accelerometer in its proper orientation. The accelerometer housing was fixed to the collar using hose clamps for the first deployment; however, issues developed with housing retention, so all future deployments were performed using high-tensile strength zip ties. We oriented the collar with the positive axis facing forward, right, and down for the *X*-axis, *Y*-axis, and *Z*-axis, respectively (Figure 1). This is similar to other research in cattle [9]; however, orientation is not always consistent across the literature [2].

### 2.2. Behavior Collection

Behavior was assigned according to an ethogram developed for extensively grazed cattle [25], including the following: grazing, walking, resting, ruminating, or grooming (Table 1 and Figure 3 and Figure 4). Two observation types were used to increase the number of behavior observations available because of a disruption in data collection caused by travel restrictions from the COVID-19 pandemic. We employed focal sampling to continuously observe individual animal behavior via two methods, including in-pasture observation and decoded video from the animal-borne camera collar system. In-pasture observation (in-pasture behavior) was conducted by trained technicians equipped with audio recorders synchronized with the atomic clock (UTC), who followed target animals and verbally noted exhibited behaviors. Audio recordings were then transcribed into a continuous dataset describing behaviors at a 1-s resolution at the corresponding time (UTC). In-pasture behavior observation occurred during regularly scheduled data collection trips, which provided 227 min of in-pasture observed behavior (Figure 5a).

The same trained technicians who conducted the in-pasture behavior observations decoded the video data collected by the animal-borne cameras. Simultaneous in-pasture behavior observations and animal-borne camera system videos were unavailable because of travel restrictions in response to the COVID-19 pandemic. Only animals with accelerometer data during this period were selected for behavior assessment (*n* = 8); 2 animals experienced equipment failure because of water infiltration into the accelerometer housing and loss of the external accelerometer in the grazing pasture. To mitigate issues of time mismatch when joining observed with accelerometer data, animal-borne camera collar observed behavior bouts were removed from the analysis if the animal switched behaviors during the 15-s video clip (*n* = 518). This provided 465 min of animal-borne camera collar video for categorizing animal behavior.

### 2.3. Data Management

Accelerometer data were visualized in Daily Diary Multiple Trace software (Wildbytes Technologies Ltd., Swansea University, Swansea, UK, version DDMT 13 Oct 2019), where adjustments were made for accelerometer start time (UTC time) and orientation on the animal, which were then exported to csv files (Figure 3). Next, behavior observations were loaded into Program R [26] and matched with time-series accelerometer features via the full_join function and plotted using the geom_line functionality available within the *ggplot* package [27]. Accelerometry signals for all behavior bouts were visually checked for time drift and removed if the accelerometer signal indicated a mismatch between expected and actual signal appearance for the observed behavior, which can occur because of misalignment with time sync, device time drift, or issues with collar fit. After building a complete dataset of verified behaviors matched with their accelerometry signatures, a category for grazing versus non-grazing was created.

### 2.4. Data Analysis and Model Fitting

Next, using the complete dataset containing both in-pasture observed behavior and animal-borne collar video behavior data, we created a series of density plots (objective 1) to assess the functional differences in discrete behaviors relative to the magnitude of acceleration on various axes using the geom_density plot in the *ggplot2* package [27,28] for both the complete behavioral ethogram and grazing versus non-grazing category (Figure 3, Figure 6 and Figure 7). To assess the appropriate pre-processing smoothing function (objective 2), we applied smoothing window sizes from 1 to 60 s to the raw accelerometer data to calculate the accelerometer features (Table 2) and extracted at 1-s epochs. The data from each smoothing window was split into training data (70%) and testing data (30%) sets and used to train a random forest from each smoothing window size using the randomforest function in the *randomforest 4.7-1.1* package [29]. Predicted outcomes from each model were created from each dataset using the predict function in base R 4.2.3 [26]. The performance of predictive models was measured using the proportion of true positives compared to false positives and false negatives using the confusionmatrix command within the *Caret 6.0-94* package [30]. We examined model performance using accuracy, precision, sensitivity, and specificity [9] from each parsimonious model from each smoothing epoch. Model accuracy represents the total number of correctly classified behavior instances divided by the total number of instances [9]. Sensitivity measures the number of correctly identified behavior instances divided by the sum of correctly and incorrectly labeled behavior instances for each discrete behavior [9]. Specificity measures the number of correctly labeled instances where the behavior did not occur, divided by the sum of incorrectly labeled behavior instances and correctly labeled instances where the behavior did not occur [9]. Precision measures the total number of correctly classified behavior instances divided by the sum of correct and incorrectly classified behavior instances [9]. These metrics of model performance were plotted against smoothing window size using the geom_point and geom_line functions in the *ggplot2 3.4.1* package [27]. See Appendix A for complete details of the R session info.

After identifying the appropriate smoothing bandwidth, we subsampled the complete dataset to identify the effect of reducing behavior observations on predictive model accuracy (*objective 3*). The data were subsampled in a structured manner to maintain the original proportion of observed behavior for each animal. A bootstrap of 100 was run for each proportion of data retained (1–100%, *n* = 190 to 28,244 observations), with data from each resampled dataset used to train a random forest classification model with methods identical to those described previously. The bootstrapped results provided a distribution of expected model fit from which the average and standard error were calculated [30]. The results were graphed by plotting averages of model performance using the geom_point and geom_line functions (Figure 8), while the geom_ribbon function was used to graph the standard deviation [27]. The results are reported in Figure 8.

The behavior composition and spatial–temporal distribution were compared between in-pasture and collar video observed behaviors. A *column* and *treemap* plot was constructed for both in-pasture- and collar video-observed behavior using the ggplot and geom_treemap functions available in the ggplot2 [27] and treemapify [31] packages (Figure 5). To compare the spatial distribution between in-pasture- and collar video-observed behavior, each observed behavior bout was matched with the nearest temporal GPS location using the nearest function in the *data.table* package [32], converted to a spatial object, and mapped according to observation method and behavior (Figure 9) using the st_as_sf and geom_sf functions in the *sf* package [33].

## 3. Results

In total, 1171, 3438, 135, and 1603 and 8285, 6580, 2524, and 4508, seconds of behavior were observed for grazing, resting, ruminating, and walking behavior for in-pasture observations and animal-borne camera collars, respectively (Figure 5). A combined total of 41,549 s (11.5 h) of animal behavior was reviewed across eight animals split among five grazing behaviors (Table 1). After visually inspecting accelerometer signals, 7324 and 6007 s of data were discarded from the in-pasture observations and animal-borne collar video observations, respectively. This left a total of 28,244 s of observation used for subsequent analysis. Grooming was only recorded for 122 s during in-pasture observation, but after signal cleaning, only 26 s remained (Figure 5). Resting was the most common behavior observed, which constituted 35.5% (*n* = 10,018 s) of the total behavior observed. Grazing closely followed, which constituted 33.5% (*n* = 9456 s). Ruminating behavior was observed for 21.5% of the total (*n* = 6106 s). Walking (*n* = 2659 s, proportion = 9.4%) represented the smallest proportion of the summed total for all observed behavior (Figure 5).

Functional differences in accelerometer signals among the discrete behaviors can be seen in Figure 6 and Figure 7. As demonstrated in both figures, grazing behavior presented the most distinct signal on the *X*-axis, indicating a transfer of acceleration force because of gravity between the *X*-axis and the *Z*-axis as the position of the head changed. Walking presented greater density on the negative *X*-axis, indicating higher acceleration values while the head was up. Finally, grooming presented the most general behavior, with a proportion of signals distributed across positive and negative values for all acceleration axes.

Model performance improved with increasing window size; the optimal smoothing window size was reached at ~10 s for all behaviors and ~15 s when discriminating only between grazing and the other behaviors (Figure 10). Walking behavior reported the lowest sensitivity across all smoothing windows, which is expected given the heterogeneous nature of accelerometer features demonstrated in the density plots (Figure 6 and Figure 7). To identify the number of observations required to achieve the desired classification model performance, we sub-sampled the data from 1 to 100% (*n* = 190 to 28,244 s) in a structured manner to maintain the original representation of each behavior. As shown in Figure 8, the behaviors with the most complex accelerometer signals, i.e., resting and ruminating, demonstrated the strongest asymptotic curves, reaching an asymptote at ~50% (n observations = 14,531). The behaviors with at least one visually unique identifier in the raw accelerometry, i.e., grazing and walking, showed less sensitivity to a reduction in the number of observed behaviors (Figure 8). Grooming had the lowest number of observed events (Figure 5) and returned the lowest classification accuracy and highest sensitivity to a reduction in the number of behaviors observed (Figure 8). After categorizing behaviors into respective activity levels, the classification accuracy results were similar, and greater than 95% accuracy was achieved using only 25% percent of the available data (Figure 8). These results demonstrate the importance of ensuring a broad representation of behaviors is observed for training classification data, while gross quantities of discrete behaviors with unique signal attributes are unnecessary.

It is important to note that walking behavior was more frequently observed in the animal-borne collar video-collected data than in-pasture observations, demonstrating an advantage to obtaining a temporally and spatially representative sample of animal behavior via the video collar (Figure 5 and Figure 9). Behaviors observed in-pasture were collected frequently throughout the day and evening, which may have skewed observations toward resting and grazing behaviors. Animal-borne collar-video-observed behaviors were distributed spatial-temporally across the pasture. They showed a spatial concentration of ruminating and walking behavior, while grazing behavior was more spatially distributed across the pasture landscape (Figure 9).

## 4. Discussion

Continuously observing cattle behavior promises to provide detailed insight into fundamental metabolic processes at the individual level. Yet, calibration models are extremely difficult to transfer among studies, requiring each study to collect a dataset of accelerometer features matched with visually observed behavior to train classification algorithms. The duration of behavior observation in training datasets ranges from less than 2 h to greater than 50 h, with most studies choosing to observe less than 10 h or more than 50 h [9]. We observed 11.5 h of behavior, which is congruent with 42% of the accelerometer studies [9]. Observed behavior is extremely difficult and time-consuming to collect, which explains the short duration of observation reported by many studies. Further, visually collected data are normally collected over relatively short periods, which may not fully represent the diversity of terrain or behaviors in the study animal.

Grazing and walking behaviors demonstrated significantly higher VeDBA values, with a greater density of higher values on the *Y*-axis compared with resting and ruminating behaviors. Grazing created a distinctly unique signature, with lower acceleration measured on the vertical axis (*Z*-axis), which was transferred to the horizontal fore and aft directions (*X*-axis), indicating the head-down posture indicative of foraging (searching and grazing) behavior (Figure 6). Our values demonstrated a greater degree of separation in the raw accelerometry signal than those reported in other free-ranging cattle using handmade accelerometry collars [20], which is indicative that substantial differences exist to create an accurate classification algorithm using the random forest machine learning architecture. Combined with a lower degree of separation and the machine learning approach to setting thresholds, this explains our higher degree of temporal and spatial accuracy compared with other studies using a threshold approach to classifying grazing cattle behavior [1,6,20].

A plethora of pre-processing methodologies have been utilized for a variety of purposes in accelerometer signal processing into various features [17]. We found that pre-processing 10-s smoothing windows and the most available data provided the best model performance, with the prediction of steer behaviors returning an accuracy of 0.98 when predicting behaviors (Figure 10a). This is in agreement with other studies in cattle [2,17] and sheep [14,34] and places our results within what is commonly expected from other similar studies [9]. Also unknown is the quantity of behavior data required to train classification models [9]. This leads many researchers either to collect as much data as possible, potentially wasting valuable resources, or utilize what is available, potentially under-sampling, which may result in unreliable models. We tested this phenomenon by proportionally reducing the number of observed behaviors and found approximately 5.5 h (50% of observed behavior) were required to obtain optimum model accuracy (Figure 8). Behaviors with distinct accelerometry signals, such as grazing (Figure 7), were the least affected by data reduction, while those with mixed signals on multiple axes, such as resting and ruminating behaviors (Figure 7), were affected most (Figure 8). Grooming, the least observed behavior (Figure 5), was the most affected by data reduction (Figure 8). This demonstrates the difficulty in predicting biologically important, but rarely observed, behaviors using accelerometry signals, which is an issue noted unilaterally within the accelerometry behavior literature [9].

The use of camera collar technology to remotely observe free-ranging animal behavior is extensively used in difficult-to-observe animals [22]. Passive camera systems allow for observing specific animal behaviors and provide ecologists with a greater perspective of species life history, including patch selection and calf survival in Caribou [35]. Further, collar video data have been utilized in training accelerometer algorithms in the Gobi Khulan (*Equus hemionus hemionus*), a free-ranging wild horse in the Mongolian steppes [18]. Our study illustrates the advantage of using animal-borne collar video cameras for behavior observation to capture a greater distribution of the animal ethogram (Figure 5), increase the spatial–temporal area over which behavior is observed (Figure 9), and from a fixed perspective, which should aid in mitigating observer bias. Most animal observations, at least in domesticated livestock, are performed continuously by at least one observer using either direct or continuous video [9,17,36].

## 5. Conclusions

Smart collars present a new frontier in animal agriculture and will improve nutrition and forage management [11] while integrating with other new technologies [12]. We demonstrated the parsimonious differences among behaviors evident in accelerometer signatures and identified optimal smoothing windows that improve the accuracy of behavior classification when using random forest machine learning methodologies. Further, we demonstrated the increased spatial and temporal diversity of behaviors collected using animal-worn cameras and compared the accuracy of behavior classification models developed using each of the two methods. While they performed with demonstrably less accuracy, the accuracy and increased quantity of behavior observed indicate this may be a viable solution for extensively managed animals whose behavior is difficult to observe.

## Figures and Tables

**Figure 1 sensors-24-03171-f001:**
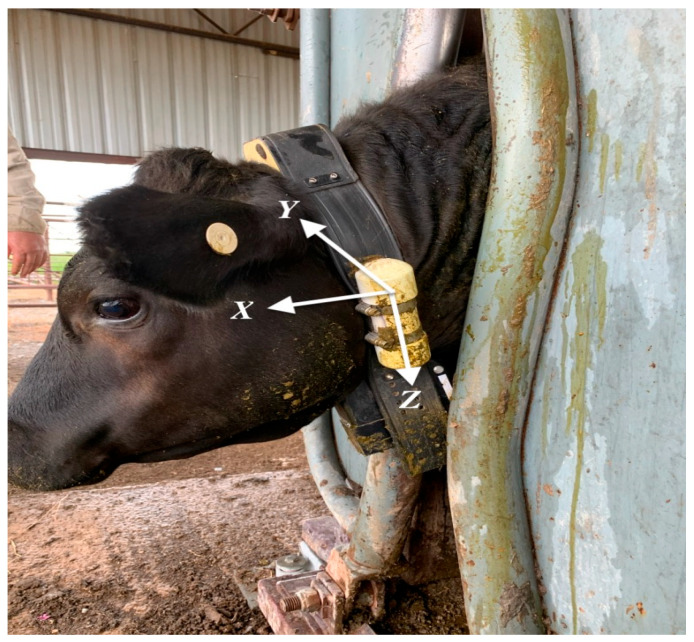
All steers were fitted with Vectronic Vertex Plus GPS collars (Vectronic Aerospace GmbH, Berlin, Germany) equipped with an accelerometer (Wildbytes Technologies Ltd., Swansea University, Swansea, UK), a GPS system, and a camera. Accelerometer housings were created using a 1-1/4 inch schedule 40 PVC pipe with a lithium-ion battery and Kaizen foam insert to hold the chip containing the accelerometer, magnetometer, and storage card. Accelerometers were changed at 3-month intervals or as needed and fastened to the left vertical strap of the tracking collar. As seen, hose clamps were initially used in the first deployment; however, after multiple failures to retain the accelerometer housing, the hose clamps were replaced with high-tensile zip ties in all subsequent deployments.

**Figure 2 sensors-24-03171-f002:**
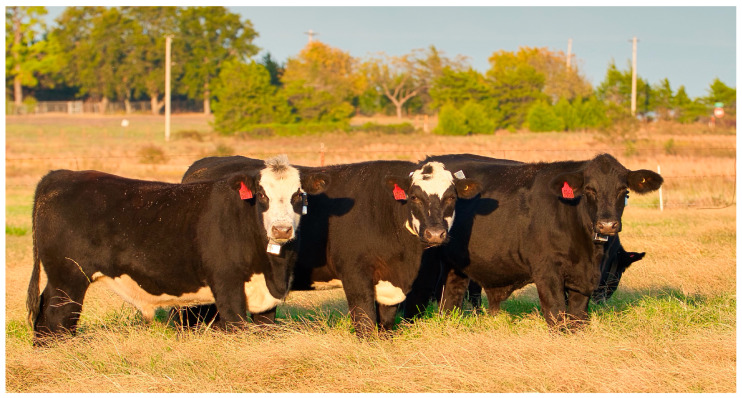
Brahman (*Bas taurus indicus*) cross steers were grazed from 21 February 2020 to 3 December 2020 in an improved grass pasture on the Pasture Development Facility owned by the Noble Research Institute. This picture demonstrates the type and quality of the steers, as well as the collar (Vectronic Aerospace GmbH, Berlin, Germany) fitment. The center steer was observed to flip his collar upside down, despite repeated efforts to tighten the collar strap. Collar spinning disorients the positioning of the accelerometer, potentially confounding accelerometry signals and mitigating the ability to classify animal behavior accurately.

**Figure 3 sensors-24-03171-f003:**
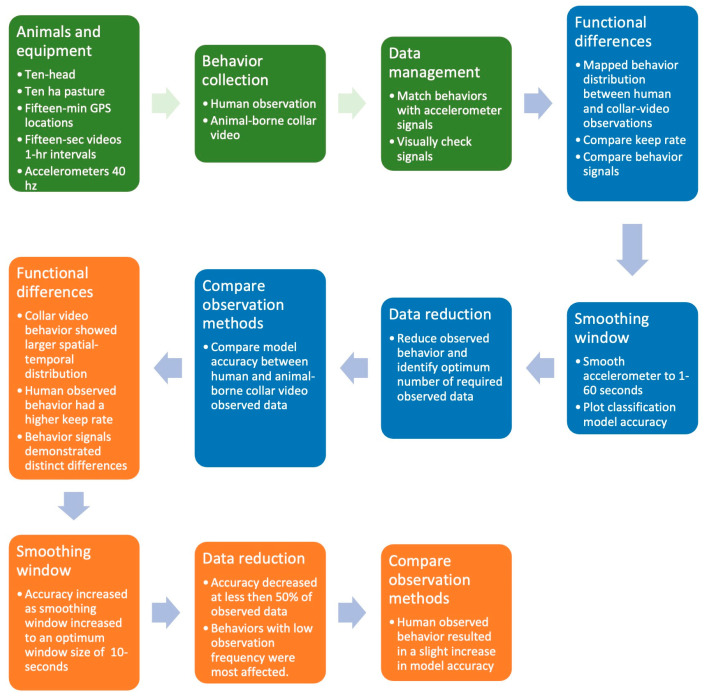
The process flow diagram describing the steps used to collect observed animal behavior and develop machine learning models to categorize accelerometer behavior.

**Figure 4 sensors-24-03171-f004:**
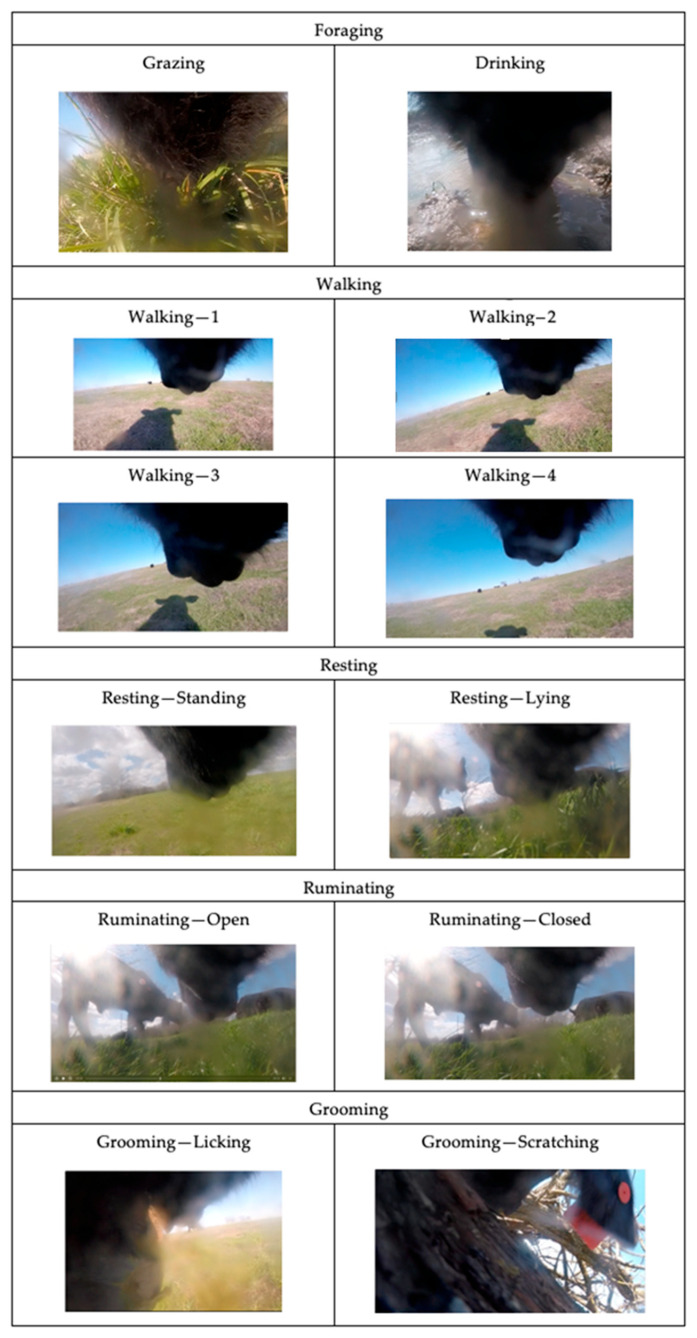
Graphical ethogram of behaviors observed using the animal-borne cameras [18]. Behaviors observed from the body parts, image tilt, and change in orientation are evident from the camera images. Foraging: Head down foraging for grass. Walking: Head up and noted head bob associated with the rise and fall of each foot, coinciding with objects growing larger within the camera frame as they draw closer, indicating forward motion. Ruminating: In the bottom jaw, a drop indicative of bolus regurgitation followed by the commencement of chewing activity. Grooming: The animal is engaging in licking or scratching activity on themselves or other animals in the social group.

**Figure 5 sensors-24-03171-f005:**
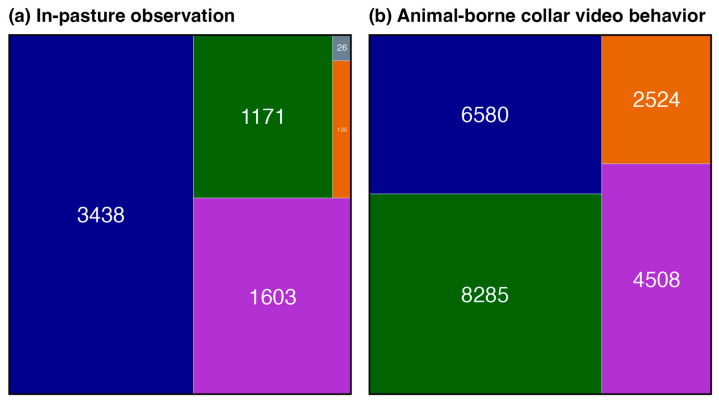
The total number of observations collected via in-pasture observation (**a**) and animal-borne collar video (**b**). This demonstrates the differences in data quality and behavior distribution among the behavior observation methods.

**Figure 6 sensors-24-03171-f006:**
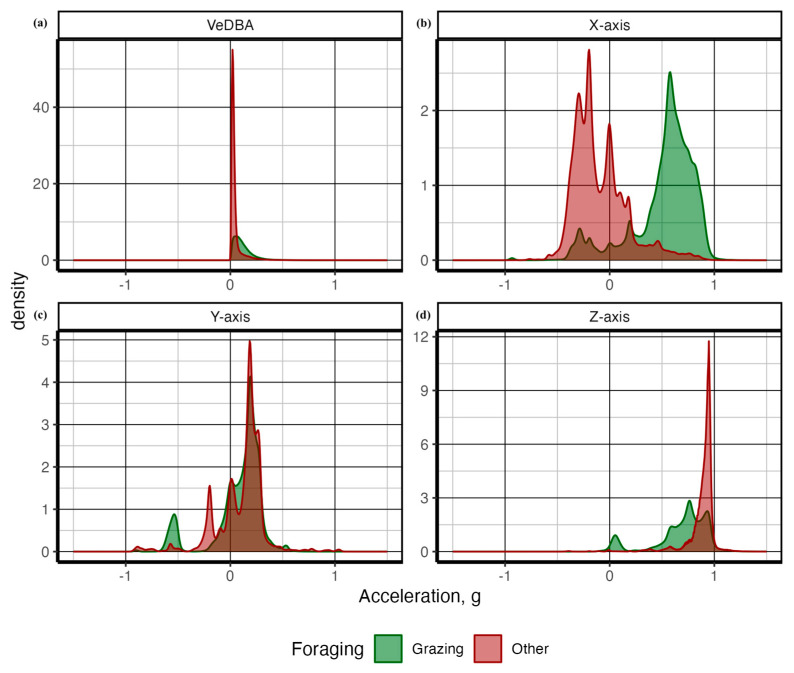
Grazing behavior indicates foraging activity and energy intake, which is fundamental to animal well-being, making it an integral indicator of metabolic status. Plotting density grazing versus other behaviors demonstrates distinctive signal separation. The head-down posture when consuming forage transfers gravitational energy from the *Z*–axis to the *X*–axis. Also, the higher energy associated with biting and tearing of forage increases VeDBA values, as seen below.

**Figure 7 sensors-24-03171-f007:**
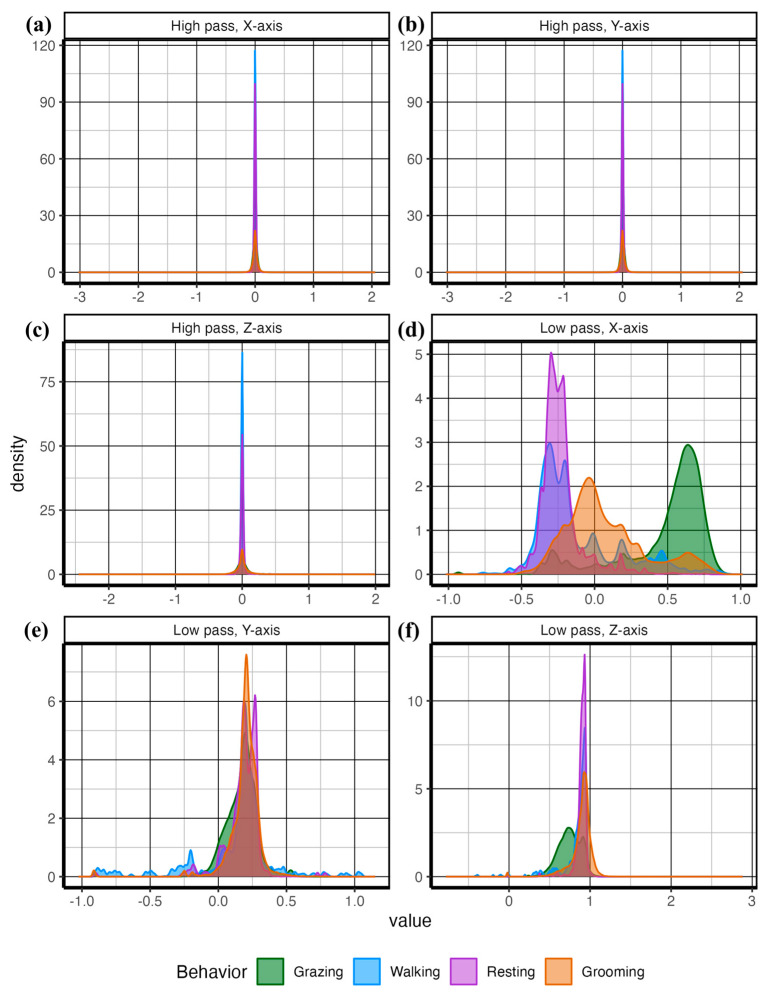
Accelerometer signals were passed through a Butterworth High- (**a**–**c**), and Low-Panel (**d–f**) pass filter. The density plot demonstrates the separation of the extracted features from accelerometer signals among the behaviors. The transfer of gravity shown as steady acceleration between the *X*-axis and *Z*-axis shows distinct separation from the other behaviors.

**Figure 8 sensors-24-03171-f008:**
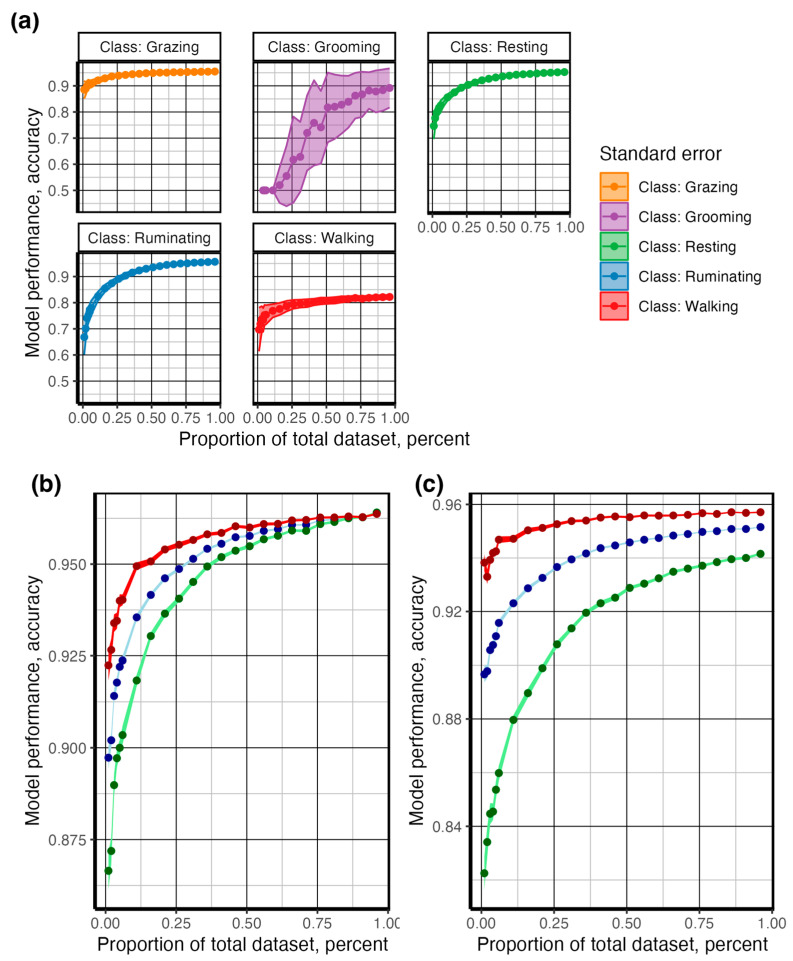
Collecting sufficient observed behavior is critical to creating a robust classification model; however, collecting more data than is required may waste valuable resources. The total data collected were split into training (70%) and testing (30%). As the total proportion of the data used in training increased, the performance of the random forest improved, as indicated by an increase in the accuracy, precision, and sensitivity metrics. (**a**) shows the effect of a structured reduction in data availability on model accuracy for the random forest classification model used to predict behavior bouts consisting of grazing, walking, resting, and ruminating. (**b**) shows the effect of structured data reduction between active (grazing and walking) and inactive (resting, ruminating, and grooming) behaviors. (**c**) shows the effect of structured data reduction when classifying grazing versus non-grazing behavior.

**Figure 9 sensors-24-03171-f009:**
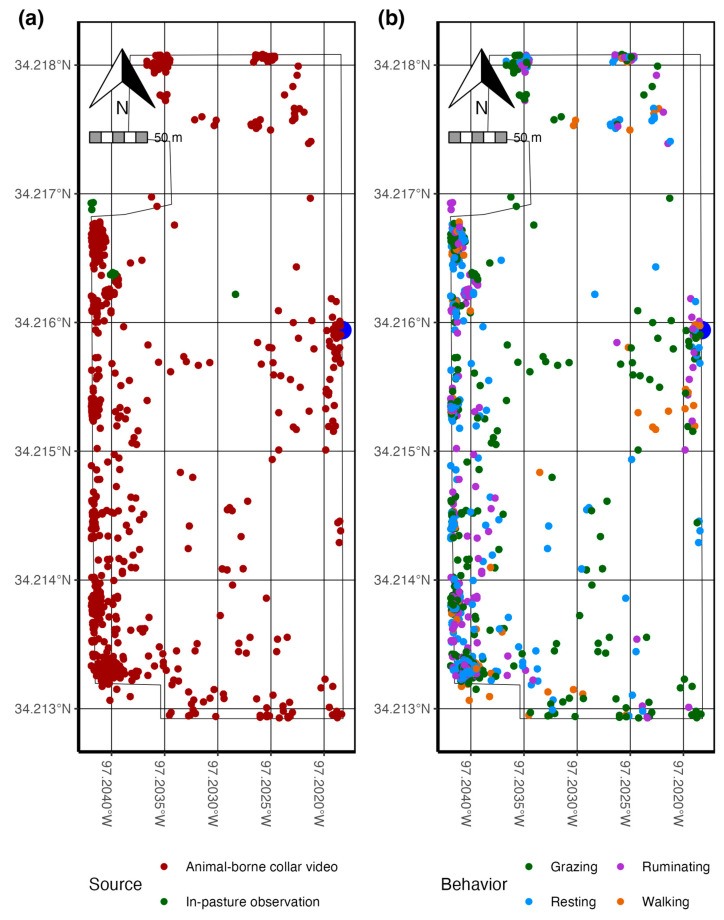
Linking behavioral observations and geolocation data indicates the spatial distribution of exhibited behaviors. This map (**a**) demonstrates the increased spatial distribution of behavior collected using an autonomous data collection system such as collar videos versus in-pasture observation. Further, this allows for an analysis of space use (**b**) for each behavior and demonstrates the advantage that animal-borne collar video behavior observation has on increasing the spatial area and temporal data collection period. This may improve model resiliency by mitigating temporal and spatial bias.

**Figure 10 sensors-24-03171-f010:**
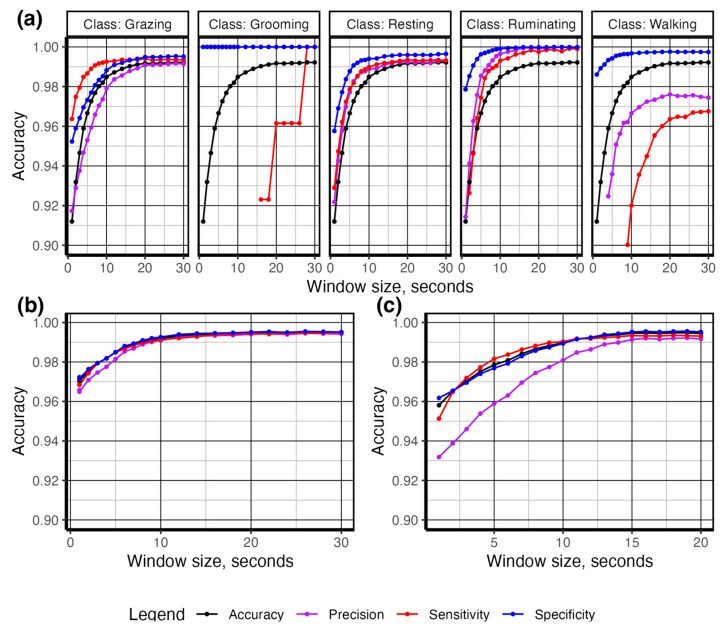
Accuracy of random forest-predicted behavior bouts. (**a**) foraging, resting, ruminating, and traveling. (**b**) active (walking and grazing) versus inactive (resting, ruminating, and grooming) behaviors. (**c**) foraging bouts (foraging vs. non-foraging) across increasing smoothing window size.

**Table 1 sensors-24-03171-t001:** Ethogram specifying the behaviors and their descriptions used to categorize observed steer behaviors. Adapted from Kilgour et al., 2012.

Behavior	Description
Grazing	Foraging for grass with head down
Walking	Minimum of 2 steps forward, with each stride initiating a 1, 2, 3, 4 pattern.
Resting	Animal idle, either standing or lying, with minimal head and foot movement.
Ruminating	The process of regurgitating a bolus, chewing, and re-swallowing, may be performed either standing or lying.
Grooming	Animal either licking, scratching, or shooing flies from themselves (personal grooming) or performing licking or scratching activity on other animals (social grooming).

**Table 2 sensors-24-03171-t002:** Features calculated from the raw accelerometer data at each smoothing window size, extracted at 1-s epochs, and used to train random forest classification algorithms to identify discrete animal behavior.

Type	Accelerometer Features
Raw acceleration	Acceleration X, Y, Z, g
Smoothed acceleration	Mean acceleration X, Y, Z, g
	Median acceleration X, Y, Z, g
	Standard deviation of acceleration X, Y, Z, g
	Minimum acceleration X, Y, Z, g
	Maximum acceleration X, Y, Z, g
Acceleration vectors	Dynamic acceleration X, Y, Z, g
	Overall dynamic body acceleration (ODBA)
	Vector of dynamic body acceleration (VeDBA)
	Mean ODBA and VeDBA
	Median ODBA and VeDBA
	Standard deviation ODBA and VeDBA
	Minimum ODBA and VeDBA
	Maximum ODBA and VeDBA
Raw magnetometry	Magnetometry X, Y, Z, geomagnetism
Smoothed magnetometry	Mean geomagnetism X, Y, Z, geomagnetism
	Median geomagnetism X, Y, Z
	Standard deviation geomagnetism X, Y, Z, geomagnetism
	Minimum geomagnetism X, Y, Z, geomagnetism
	Maximum geomagnetism X, Y, Z, geomagnetism

## Data Availability

Data available from authors upon reasonable request.

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
