# Peer review of "Machine Learning Methods and Visual Observations to Categorize Behavior of Grazing Cattle Using Accelerometer Signals"

_sensors, 2024, doi:10.3390/s24103171_

Round 1
Reviewer 1 Report
Comments and Suggestions for Authors
The authors presented an extensive analysis of data collected by the use of a tri-axial accelerometer. Combining these data with two means of observation, direct and video, is an original approach. However, behavioral sampling and event characterization are unclear to understand the results. I think the manuscript can be improved, I detail major and minor concerns below.
Major
Considering the study carried out, I see a validation study, it is necessary to be more specific in the vocabulary to understand what was done. When Authors write about a behavioral evaluation method, they are actually referring to technological media of evaluation (video recording and notes by direct observation).
I see that a clear description of the sampling methods is missing. I suggest considering the guide from Martin & Bateson, 2013. I believe that the authors applied a focal sampling rule, recording data for each individual. In addition, on recording rule, Authors apparently recorded data at fixed intervals at two different periods (September-October & February-May). It is important that readers know how the observation session was divided into a different time sampling. This will allow us, I mean as readers, to understand how the accelerometer data were simultaneously processed to interpret the observed behavior.
Both methodological media (direct observation & video observation) are useful for validation. Figure 4/a is not clear. From this section I couldn't really follow how the data was processed. For example, lines 170-171, what behavioral data was compared to the accelerometer output. I cannot find information on the number of behavioral events evaluated, and thus understand what pattern of behavioral occurrence was revealed by the video or by direct observations.
The authors suggest a series of studies as technical references over the M & M section. However, since the present work is really a validation, further explanation of the procedures is required. According to Figure 3, the authors have done much of what I hope to see. But it is not clearly explained in M&M or it appears in results without methodological support. For example, the parameters of accuracy, precision, sensitivity and specificity have to be clearly defined earlier. Besides, some lines like 393-395 would be useful in the M & M section, rather than in the result section. Generally speaking, perhaps, figure 3 is a good guide to redone the M & M section, describing precisely what Authors performed from behavioral data collection and data analyses. On this point, I remember lines 64-67 in the introduction. Perhaps researchers applying machine learning methods may offer a better explanation of what they have done for readers, particularly, ethologists and future technicians in the farm.
Particularly, considering the title of the manuscript and the state of the art, perhaps it would be more useful to see in the result section a VeDBA profile combined with the x, y, z profiles of the accelerometer during the behavioral events evaluated over the 5 hours. Perhaps this could replace or be shown before figures 6 & 7.
Due to the lack of information and/or the lack of order in the presentation, the objections indicated above do not allow following result presentation. Therefore, the discussion is not supported by the previous sections or by the present text developed by the authors.
Minor
Abstract
Lines 12-13: behavioral classification may not be essential to identify metabolic energy and health status, since monitoring activity without discriminating behaviors would be enough. Considering, for example, Brown et al 2013, as revision https://doi.org/10.1186/2050-3385-1-20. So, I suggest modifying the sentence.
Introduction
Lines 39-54, I have 2 issues here that the authors should take into account: a) on sustainable use of farm species, the real usefulness of the sensors and the classification of behavior as a research question needs further arguments. First, I understand and share the authors' approach. But I suggest that the authors should be more explicit or precise when they are talking about the usefulness of measuring behaviors with sensors. What is really the advantage for the farmer applying this technology? Second, it seems that the classification of behavior would not be a common question in research. I think about this meaning. I believe that the classification is a validation, very useful, but it is not in itself a question. Therefore, I suggest modifying both topics. Besides, please try to be specific, considering line 55. I suggest: Monitoring animal behavior by sensor is an extremely difficult endeavor.
Lines 64-78, on this paragraph, I have doubts about the focus of this paragraph. Innovative development always challenges researchers, which is not the problem with innovation. Perhaps the focus should be on the advantages and disadvantages of multidisciplinary approach, perhaps many ethologists still lack training to make use/understand multiple advances in computing. So, I suggest that the paragraph be revised.
Lines 79-84, validations are always laborious for sure. But the effort and time dedicated gives robustness to the conclusions. This section should be removed, as it seems to encourage not performing validations, or it should be rewritten. Next, I suggest focusing on domestic species. In fact, there are studies on the subject. For example, Huck & Watson https://doi.org/10.1016/j.applanim.2019.04.016. I suggest reviewing this paragraph, I offer greater precision and/or better search in the literature.
Other issues, that seem minor, a) on animals and equipment: What was the weight of the collar, accelerometer, GPS? What is the relationship between the weight of the device and the average or estimated animal body weight? This information is important to see if the technical requirements were reached; b) the statistic is not described in M&M. There is information in the table titles, but it is not enough to understand the procedures. The following must be reported: unit of analysis, factor structure of source of variation, statistical method; c) please review the expression "human observation". Both human observation and video observation were done by humans. Perhaps direct observation is more appropriate instead of human observation.
Reviewer 2 Report
Comments and Suggestions for Authors
The objective of this study was to identify accelerometer signal separation among parsimonious behaviors of grazing cattle. The workload of this study is significant.
However, the manuscript did not present in a very clear and coherent way.
What is the relationship between human observation and video captured data? As far as I understand, both them are for the ground truth validation, video based data could provide more data to validate the result. But the author made many comparisons between them. I don't know why the author do this. In addition, the author did not mention how the video captured data was processed. Is it by a computer vision analysis? or By human watching and labeling?
Line 275, the author conclude that "the optimal smoothing window size was reached at ~10 seconds for all behavior, and ~15-seconds when discriminating only between grazing and other behavior (Fig 9).". There no detailed introduction on the methods. I wonder how the results was achieved.
Line 248, Is there typo in "between four grazing behaviors (Table 1)." ? There are five behaviours in Table 1.
In Figure 4, the clean and unclean icon is incorrect.
Figure 6 is reversed.
Figure 8, why ruminating is not metioned.
Comments on the Quality of English Languageno comments.
Round 2
Reviewer 1 Report
Comments and Suggestions for Authors
I carefully read the revised version and I saw that Authors improved the manuscript. But I still found some details that must be improved. Honestly, I don´t think that this version is ready to be published.
Title
In order to be more specific, why don´t you add “machine learning" before “Methods to categorize….”? Otherwise, it is unspecific. For example, you may apply a manual method to categorize accelerometer data. So, I strongly recommend being more specific. In fact, former title was well “Machine learning and behavior observation approaches to categorize extensive grazing behavior using collar borne accelerometers''
Abstract
Considering the work done by Authors, lines 12-13 (first sentence) is speculative. There is no information supporting inference about metabolic energy and space use, so, I strongly suggest eliminating this sentence. In addition, please try to avoid nonspecific terms such as “animals”, you know fish as well as cows are animals, and accelerometers don ́t fit at all. Check it over the manuscript.
I found really good the objectives expressed, 1 to 4. But, the number 2 is not clear enough. Please revise it. I feel again that there is a lack of specificity.
It seems that lines 31-33 refer to the accelerometer, right? Then, lines 33-35 I found a statement about the comparison of observation by the camera on the steer collar and human observation. This is not clearly justified, and I think it is not comparable considering the sampling effort (10 d vs 2). Perhaps it is obvious that human observation would be more accurate and precise than the other type of observation. At the end of the section, I have another question, how accurate was the analysis of the accelerometer data with respect to video analyses?
Introduction
Lines 56-57, this statement is not clear for this context. Perhaps, it could be eliminated. I found that animal-borne sensors and accelerometers are the key words for this paragraph. Please revise.
M&M
I notice that the M&M section has improved. Thanks for the effort made. However, the clarity in the text raises new questions. Why did they use two types of observations? human observation vs video observation. There is no explanation. Considering the study period (10 d for video), it may be appropriate to analyze the accelerometer data with respect to this methodology, since Authors have a lot of data to analyse with.
Once again, I found figure 3 not very useful. For example, in the second box a major problem is observed, the authors obtained the same number of hours of observation employing the same sampling rule (4 sampling point per hour) during different periods of time (human observation= 2d and collar video=10). This is an error o it is not clear.
So, as I previously suggested, each box could be a section of M&M. Each box would be subtitle with its description in detail. Therefore, I suggest reviewing this, perhaps eliminating this figure
Table 1 needs to clarify whether the ethogram was useful for both types of observations, human observation and video recordings (and further behavioral analyses).
Although I have doubts that this ethogram has been useful for both types of observations. For example, how did they detect Walking if, through the camera on the collar, they could not verify the steps taken by the animal? This requires an explanation, or perhaps the development of another ethogram.
Results and discussion
Unfortunately, I cannot analyze these sections due to the lack of information in M&M or the complex organization of this section.
Reviewer 2 Report
Comments and Suggestions for Authors
There are still couple of major issues in this manuscript. 1) There is less value to study the objective 1 and 4. The result with 10s epoch from objective 2 is common and no innovation. The result of objective 3 is not clear and has less reference significance for the other research. 2) Compare the method of human and collar video observations has less meaning. Figure 3 doesnot present the most attractive points. 3) RF was used a lot. The author choose RF due to its good performance is untenable. 4) It is a mess of refering figures in the manuscript.
There are two figure 1.
Line 296,Fig 5 should be Fig 6?
What is the volume of the Lithium battery.According to the function of these sensors,the data collection is power consuming with 40Hz of acceleration collection, in addition to video data collection.
Figure 5., what is the legend for Foraging? No introduction of foraging in Table 1. Why there are three colours in Figure 5?
Line 277-278, Results are reported in Fig8. Fig8 is not here. It is a mess in the manuscript while refering figures.
Figure 7, 321-326, duplicated introduction of the concept the sensitivity, accuracy, specificity.
Figure 6. no ruminating behaviour. It seems that grooming was lest behaviour detected which was introduced in line 307 and 308.
In Figure 3., Mapped behaviour distribution between human and collar video observations. What is the significance of this comparison.
In Figure 3., Compare observation method between human and collar video observations. Having no meaning to do that. As far as my understanding, both of these two method were useful to increase data volume and variaty. It is hard to say that which method is superior to the other.
In Figure 3., Human observed behaviour resulted in a slight increase in model accuray, How does the author explain this? Is this due to the data was collected by Human?
Add arrows in Figure 3 to let the readers understand the data flow direction.
Line 232, window sizes from 1 to 60 seconds to the raw accelerometer data to calculate the accelerometer features. I think the author only make comparisons with which epoch achieved the best results. It is not smoothing window size.
Line 149 -169 has been discripted in fig1 and fig2.
Line 447-449, We tested this phenomenon by proportionally reducing the number of observed behaviors, and found that at least 50% (observation hours < 5.5 hours) of our observed behavior data were required to obtain optimum model accuracy (Fig 8). What is the meaning of the least 50% of our observed behavior data? Could the author propose a least observation hours for the other reseachers to refere?
Figure 9.,It seems the behaviour data only from collar-videos versus human observers. It was suggested to utilize the predicted behavour data from the RF model.
Round 3
Reviewer 1 Report
Comments and Suggestions for Authors
lines 24-30, I suggest eliminating them since it is not clear yet why Authors used 2 behavioral methods for measuring behaviours. Furthermore, as I earlier said, different sampling was applied, which generates more doubts than certainties. It makes noise, so I think that Authors could focus on results. Besides, avoid clarifications such as (e.g. 10 vs 2 days), and add a conclusion to the accelerometer data processing method. This is more important than informing that pasture observation was more accurate than collar video observation.
I have revised this manuscript for the third time, from my point of view as an ethologist, I see that the introduction has improved a lot, and I can easily reach the specific objectives. Congratulations, and thanks for your work. However, materials and methods still present problems to be publishable study. In particular, I am concerned about the use of 2 observation methodologies; I am not satisfied with the reply in the cover letter (response 10).
First, returning to the introduction, lines 117-119 confuse the reader since the Authors did not obtain simultaneous data with the different methodologies, therefore, they cannot make direct comparisons between observations in pastures and animal-borne video cameras. This is based on data from M&M (Sep-Oct 2020 and Feb-Mar 2020). Besides, this not allow Authors reach specific objective 4 (lines 131-133); this objective must be deleted. Second, on my recommendation about 2 ethograms, I understand that it may be simple for you to watch the video and deduce that the animal was walking. But the observation methodology does not work as you pointed out. Furthermore, that deduction associated to watchting video may be different for another observer. This is why the ethogram exists, and must be specific. For example, I wonder how you deduce the animal's strides in a 1,2, 3,4 pattern when you don't see the feet in the video? This is incorrect. It must be changed. An ethogram is needed for the video.
Finally, the author's response 6 helps to understand the methodological strategy. Please add this explanation in materials and methods.
Reviewer 2 Report
Comments and Suggestions for Authors
All the specific comments from me has been addressed properly. Thanks.
Author Response
Thank you for your effort in reviewing this manuscript